# Increased warm water intrusions could cause mass loss in East Antarctica during the next 200 years

James R. Jordan [1,2] ✉, B. W. J. Miles [3,4], G. H. Gudmundsson [1], S. S. R. Jamieson [3], A. Jenkins [1] & C. R. Stokes [3]

The East Antarctic Ice Sheet (EAIS) is currently surrounded by relatively cool water, but climatic shifts have the potential to increase basal melting via intrusions of warm modified Circumpolar Deep Water (mCDW) onto the continental shelf. Here we use an ice sheet model to show that under the current ocean regime, with only limited intrusions of mCDW, the EAIS will likely gain mass over the next 200 years due to the increased precipitation from a warming atmosphere outweighing increased ice discharge due to ice-shelf melting. However, if the ocean regime were to become dominated by greater mCDW intrusions, the EAIS would have a negative mass balance, contributing up to 48 mm of SLE over this time period. Our modelling finds George V Land to be particularly at risk to increased ocean induced melting. With warmer oceans, we also find that a mid range RCP4.5 emissions scenario is likely to result in a more negative mass balance than a high RCP8.5 emissions scenario, as the relative difference between increased precipitation due to a warming atmosphere and increased ice discharge due to a warming ocean is more negative in the mid range RCP4.5 emission scenario.

The East Antarctic Ice Sheet (EAIS) is the single largest potential contributor to future global mean sea level rise, containing 52.2 m of sea level equivalent (SLE)[1]. Despite this large potential for mass loss, most estimates of its recent mass balance give close to zero (e.g. the latest IMBIE assessment[2], 5 ± 46 Gt a[−1] from 1992 to 2017), but some suggest large gains (e.g. Gardner et al., 2018[3], +61 ± 73 Gt a[−1] from 2008 to 2015) and others suggest large losses (e.g. Rignot et al., 2019[4], −51 ± 13 Gt a[−1] from 1979 to 2017), e.g. Fig. 1. However, although overall values of mass balance differ, most studies identify mass loss in the Wilkes Land region, especially drainage basin C–D (Fig. 1).

The ISMIP6 project[5] found that, whilst models consistently predict a negative mass balance for the West Antarctic Ice Sheet in the period leading up to 2100, there is a much greater disagreement amongst models about whether the EAIS will have a positive or negative mass balance over the same period[6,7]. The most recent simulations show a general trend for future East Antarctic mass balance to be accumulation dominated, with increasing emission scenarios leading to a gain in mass, although this is not found for all models, particularly when using low emission scenarios[8,9]. A warming climate is expected to lead to greater precipitation over Antarctica[10] due to the greater moisture-holding ability of warmer air, and hence a positive change in mass balance unless there is a dramatic increase in ice discharge to compensate.

Ice sheets contribute to sea level rise via surface melting and via ice discharge over the grounding line, a process enhanced by ocean-driven melting of floating ice reducing the buttressing effect of ice shelves[11,12]. A warming climate is likely to affect the ocean temperature surrounding the EAIS via indirect means, with the strongest observed warming trends currently occurring in water to the north of ice shelf cavities rather than in their direct vicinity[13,14]. However, there remains

[1]Department of Geography and Environmental Sciences, Faculty of Engineering and Environment, Northumbria University, Newcastle upon Tyne, UK. [2]Laboratoire de Glaciologie, Université libre de Bruxelles (ULB), Brussels, Belgium. [3]Department of Geography, Durham University, Durham DH1 3LE, UK. [4]School of Geosciences, University of Edinburgh, Edinburgh, UK. ✉e-mail: James.Rowan.Jordan@ulb.be

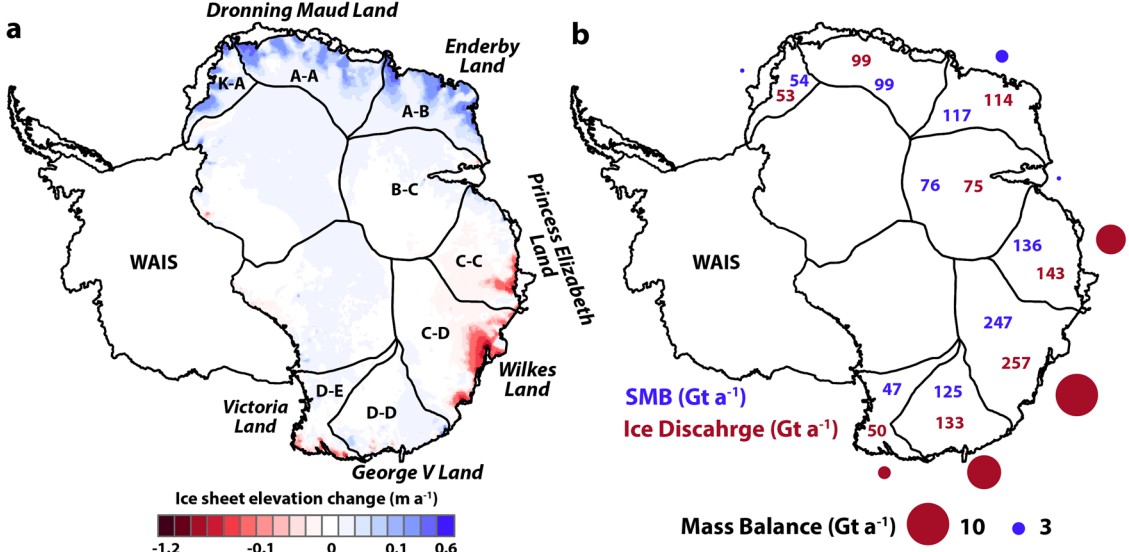

**Fig. 1 | The current state of East Antarctic mass balance.** 2003–2019 East Antarctic annual elevation change (**a**), adapted from Smith et al.[54]. 1979–2017 East Antarctic mass balance (**b**), with surface mass balance (blue numbers), ice discharge (red numbers) and mass balance (circles, red for negative and blue for positive, scaled by area) adapted from Rignot et. al.[4]. Drainage basins from Zwally et al.[49] are overlain, with data shown for the eight drainage basin of interest for this study.

the potential for substantial future warming due to climatic conditions forcing a shift in the particular water mass driving ice shelf melt[15].

At present, the primary water mass in contact with East Antarctic ice shelves is relatively cool (≈−1.5 °C), but recent observations show intrusions of warmer (≈0 °C) Circumpolar Deep Water (CDW) in the area surrounding Totten Glacier[16,17], Vincennes Bay[18], Shirase[19] and Amery ice shelf[20]. The Antarctic Slope Front (ASF) has been shown to regulate these intrusions, with a strong ASF limiting cross-shelf intrusions whilst a weaker ASF leads to more intrusions[21]. The strengthening (positive phase) and weakening (negative phase) of the band of westerly winds around Antarctica is known as the Southern Annular Mode (SAM). A positive SAM has been shown to weaken the ASF via wind-sea surface interactions, allowing increased intrusion of CDW over the continental shelf[22]. Observations of the SAM have shown a positive trend since the 1970s, with this trend predicted to continue with a warming climate[15,23]. Numerical modelling studies have also shown that the position of coastal currents and the upwelling of warm CDW would shift southward with a more positive SAM[24,25]. A poleward shift in the southern boundary of the Antarctic Circumpolar Current has been observed and modelled over the last 30 years off East Antarctica, potentially leading to warming shelf water[26]. In addition, ocean water freshening due to increased ice-sheet melt-water production has been shown to enhance CDW transport towards the continental shelf via a strengthening of the surface stratification near the Antarctic coast[27], causing a positive feedback loop with increased basal melting[28]. A previous study found there to be spatial variation in this freshwater feedback loop, with the Western Antarctic region seeing a cooling of ocean temperatures and the Ross Sea region in East Antarctica a warming in ocean temperatures[29]. A comparison of two different coupled models found one undergoing oceanic warming whilst the other experienced cooling due to freshwater feedbacks[30]. Recent ocean observations, however, have noted a poleward shift in CDW in East Antarctica since 2010[15]. Paleoenvironmental and paleoceanographic records from the Wilkes Subglacial Basin also find evidence for past grounding line retreat coinciding with periods of increased oceanic warming during late Pleistocene interglacials, most likely a result of CDW intrusions[31]. Evidence for a past retreat of the grounding line in the Wilkes Subglacial Basin of the order of 700 km has also been shown to correspond to ocean temperatures 1–2 °C warmer than present conditions, equivalent to the relative difference between shelf

water and CDW[32]. If changing climatic conditions lead to CDW intrusions becoming more common in the future, as predicted by Herraiz-Borreguero and Garabato[15], East Antarctic melt rates could grow to resemble the far higher CDW driven melt rates seen in West Antarctic ice shelves like Pine Island Glacier[33] and Thwaites Glacier[34].

Here, we seek to quantify the sensitivity of EAIS mass balance over the next 200 years to a shift to more frequent CDW intrusions driven by a change in ocean circulation when compared to a future emissions scenario with no change in large scale ocean dynamics. We are primarily concerned with determining the effect such a shift would have upon the future mass balance of the EAIS rather than explicitly simulating the mechanisms by which these intrusions would occur, thus seeking to ascertain the spatial and temporal impact of such a switch across the EAIS. We consider the effect of both an intermediate (Representative Concentration Pathway, RCP-4.5 emissions scenario) and extreme (RCP-8.5 emissions scenario) warming scenario on both ocean and atmospheric forcing, as they have both been shown to produce conditions likely to lead to increased intrusions of CDW over the continental shelf[25]. To achieve this, we use a numerical ice sheet model driven by parameterised atmospheric and oceanic forcings to determine the future mass balance of the EAIS. This represents the first study to examine the potential impact such a shift would have on the EAIS as well as identify the particular regions that are most at risk to rapidly increasing oceanic melting.

## Results

We simulate the effects of both a continuation of the present day ocean regime and a shift towards an ocean regime dominated by CDW intrusions combined with an intermediate (RCP4.5) and extreme (RCP8.5) emissions scenario). We define the current, relatively cool, oceanic regime from here on as Shelf Water forcing (SW)[35] and the potential, relatively warm, oceanic regime that has become dominated by CDW intrusions mixing with shelf waters as modified Circumpolar Deep Water forcing (mCDW). We also assume, for the sake of simplicity and ease of comparison between different forcing regimes, that the general pattern of future atmospheric and oceanic forcing over the next 200 years will be an increase to a maximum value followed by a period of relatively constant forcing at this new maximum. As such, we apply these forcings in a manner that increases linearly from present conditions to a maximum after 100 years with these maximum values

held constant for a further 100 years. This allows us to investigate the response of the EAIS to both increasing atmospheric and oceanic warming as well as a period of committed atmospheric and ocean warming[36]. A full description of our numerical modelling framework as well as the atmospheric and oceanic forcings applied can be found in the Methods section.

We simulate four distinct climate scenarios, namely; (1) RCP4.5 atmospheric and oceanic warming, with ocean properties representative of a cooler SW ocean regime (RCP4.5-SW); (2) RCP8.5 atmospheric and oceanic warming, with ocean properties representative of SW (RCP8.5-SW); (3) RCP4.5 atmospheric and oceanic warming, with ocean properties representative of a warmer mCDW regime (RCP4.5-mCDW) and (4) RCP8.5 atmospheric and oceanic warming, with ocean properties representative of mCDW RCP8.5-mCDW. To account for model uncertainty associated with parameters for which no observational constraint exists results are given relative to a reference run with present day forcing held constant for 200 years.

We first consider the effect of how ice volume in the domain changes throughout the model simulations by considering both the total ice volume in the domain (Fig. 2a, b) as well as the Volume Above Flotation (VAF, Fig. 2c, d) and the floating ice (Fig. 2e, f). VAF is the component of an ice sheet that is fully grounded and its reduction in mass directly contributes to sea level rise, whilst a reduction in the floating component of an ice sheet does not contribute to sea level rise, as the ice is already displacing an equivalent amount of water. Note that when we refer to total ice volume throughout this manuscript we are referring to the combination of both VAF and floating ice. We also show the grounded area of ice in contact with the bed, with a reduction in grounded area being a good representation of grounding line retreat whilst an increase in grounded area implies grounding line advance (Fig. 2g, h). These values are summarised in Table 1 and an in depth look at how these properties change on a region by region basis is shown in the supplementary material.

Total ice volume relative to the reference run (Fig. 2a) for the SW scenarios show an overall increase in total ice volume at the end of model simulations, with the RCP8.5-SW scenario showing a greater level of increase of +13,859 Gt than the RCP4.5-SW scenarios increase of +1825 Gt. The mCDW scenarios, in contrast, show a negative trend in ice volume that is of greater magnitude than the positive trend seen in the SW scenarios. As before, the RCP8.5 scenario shows a smaller reduction of −75,232 Gt of ice volume than the RCP4.5 scenarios reduction of −91,861 Gt. The rate of change of ice volume relative to the reference run (Fig. 2b) is fairly constant for the SW forced scenarios, whilst for the mCDW forced scenarios it reaches a maximum rate of mass loss after 50 years, 50 years before the maximum forcing is obtained.

Total VAF relative to the reference run (Fig. 2c) increases in both the SW forced scenarios and the RCP8.5-mCDW scenario for the first 100 years of model simulations. The SW forced scenarios continue this positive trend for the following 100 years, with total VAF increases of +26,885 Gt (−74 mm SLE) for the RCP8.5 scenario and +12,120 Gt (−33 mm SLE) for the RCP4.5 scenario by the end of the simulation, respectively. The RCP8.5-mCDW scenario reaches a maximum of +4573 Gt increase in VAF (−13 mm SLE) after 110 years before beginning a negative trend in VAF to a final reduction of −1398 Gt (a cumulative +4 mm SLE) at the end of the simulation. Maximum precipitation (Fig. S5, supplementary material) coincides with the beginning of the maximum forcing period (100 years into the simulation) whilst maximum ice discharge occurs midway through the second half of the model simulations. The RCP4.5-mCDW scenario shows a negative trend in VAF throughout the entire simulation, reaching a final value of −17421 Gt (+48 mm SLE) by the end of the simulation. The rate of change of VAF relative to the reference run (Fig. 2d) shows that the greatest magnitude of VAF change does not coincide with the

imposition of maximum forcing after 100 years. All scenarios, even those with a positive rate of VAF change, show a more negative trend in VAF change in the second half of the simulations, despite constant forcings. This illustrates there is a delay in the response of the ice sheet to maximum oceanic melt forcing, with the rate of ice discharge increasing after maximum melting has occurred.

Total floating ice relative to the reference run (Fig. 2e) shows a negative trend in all simulations, albeit more pronounced in the mCDW forced scenarios. At the end of the model simulation the RCP4.5-SW has experienced a −10,295 Gt change in floating ice, the RCP8.5-SW a −13,026 Gt change in floating ice, the RCP4.5-mCDW a −73,834 Gt change in floating ice and the RCP8.5-mCDW a −74,440 Gt change in floating ice. Even when the ice sheet in a given scenario is gaining total ice volume or VAF, it is still losing floating ice over the course of the model simulations, representing a shift in the distribution of mass in the system from floating ice shelves to inland ice due to increased precipitation. In all scenarios there is a greater loss of floating ice than VAF due to floating ice being in direct contact with a warming ocean. The rate of change of floating ice (Fig. 2f) shows a fairly consistent rate of mass loss for the SW forced scenarios whilst the mCDW forced scenarios reach their greatest rate of mass loss after 50 years. This rate is then maintained for the next 50 years before reducing in magnitude for the second half of the model simulation.

Grounded area (Fig. 2d) is found to decrease in all scenarios, with the reduction in floating ice leading to less buttressed ice shelves and a corresponding increase in ice discharge and retreat of grounding lines. The greatest loss in grounded area is seen in the mCDW forced scenarios, which lose −44121 km² for the RCP8.5-mCDW and −47174 km² for the RCP4.5-mCDW scenario. The SW forced scenarios have losses in grounded area equal to −3835 km² for the RCP8.5-SW and −4701 km² for the RCP4.5-SW scenarios. The rate of change of grounded area (Fig. 2h) reaches a maximum rate after 100 years of the model simulation and maintains this rate for the rest of the simulation, albeit with a lot of noise (caused by the fact that in our model framework a cell can be either grounded or ungrounded but not a mixture of both, causing step changes in grounded area when an element ungrounds).

After 100 years of simulation time, the model reaches maximum forcing, with both atmospheric and oceanic forcing increasing linearly since the simulation began, causing spatial variation in the change of ice discharge and surface precipitation (Fig. 3, with the values summarised in Table 2). The RCP4.5-SW scenario (Fig. 3a) and RCP8.5-SW scenario (Fig. 3b) both have a positive overall mass balance after 100 years of +80 Gt a⁻¹ (-0.2 mm a⁻¹ SLE) and +181 Gt a⁻¹ (−0.5 mm a⁻¹ SLE), respectively. All individual regions have a positive mass balance, with the exception of the D−D region in the RCP4.5-SW scenario which is approximately in balance. The RCP4.5-mCDW scenario (Fig. 3c) is the only scenario with an overall negative mass balance of −65 Gt a⁻¹ (+0.2 mm a⁻¹ SLE). All regions except A−B, C−C and D−E have a negative mass balance, with D−D having the most negative at −34 Gt a⁻¹. While the RCP8.5-mCDW (Fig. 3d) scenario has a positive overall mass balance of +45 Gt a⁻¹ (−0.1 mm a⁻¹ SLE) after 100 years, four regions, K−A, A−A, B−C and D−D, have a negative mass balance. A shift to mCDW forced melting has the potential to increase ice discharge by a large amount (in the case of the Amery basin by up to 800%, see Table 2), although this is not uniform in magnitude or space, with a large amount of regional variance. A summary of these results is given in Table 2, with in-depth results on a regional basis shown in the supplementary material).

When the maximum atmospheric and oceanic forcing that was reached after 100 years of model simulation is maintained for another 100 years (representing the period 2100−2200) there is time for the ice velocities to adjust to the increased forcing, generating ice discharges that are higher when compared to those after 100 years whilst surface precipitation remains constant (Fig. 4, with the values summarised in Table 3). As before, the RCP4.5-SW

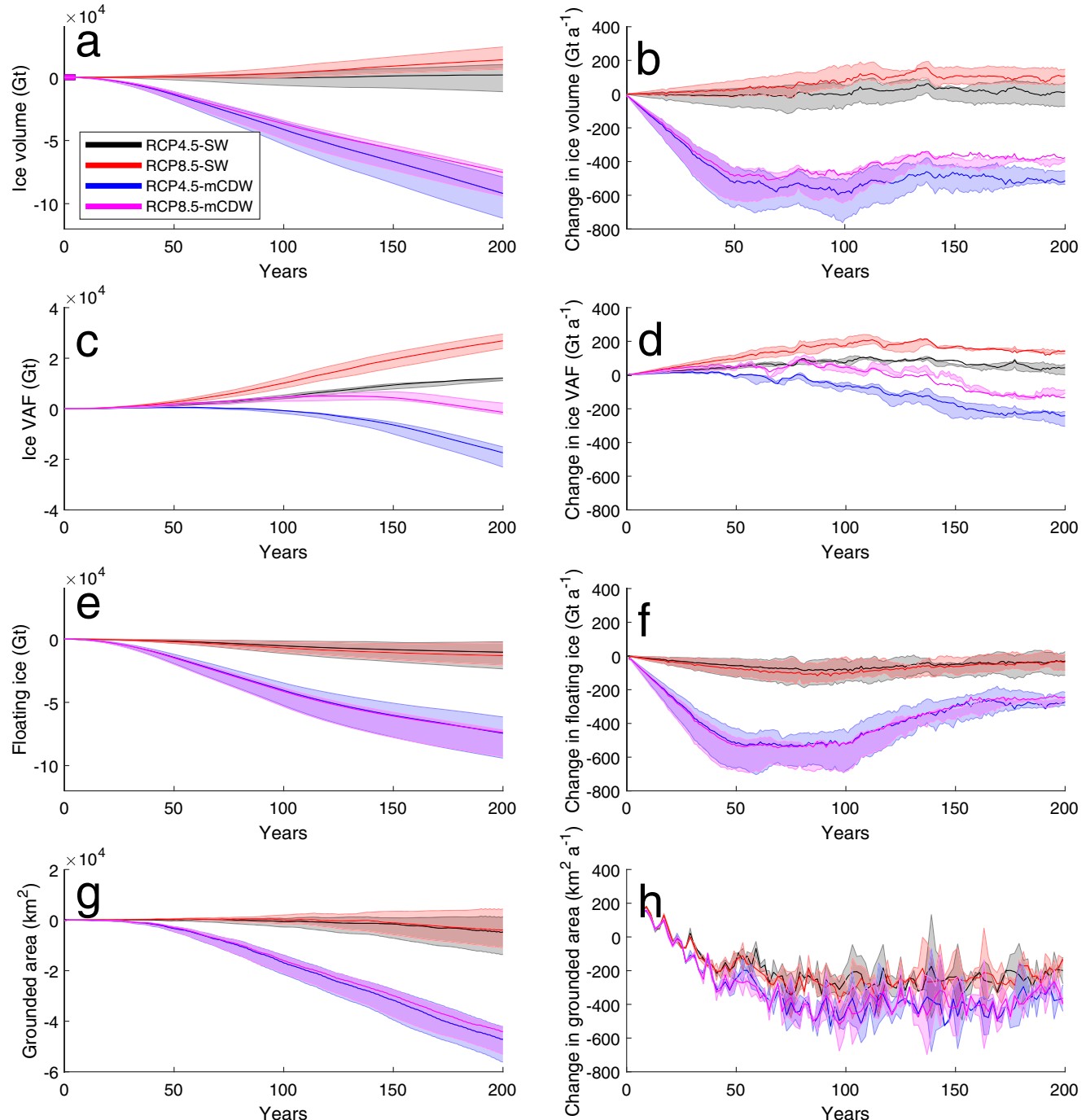

**Fig. 2 | The East Antarctic Ice Sheet over the next 200 years.** Total (**a**) and rate of change (**b**) of ice volume, total (**c**) and rate of change (**d**) of ice volume above flotation (VAF), total (**e**) and rate of change (**f**) of floating ice, and total (**g**) and rate of change (**h**) of grounded area time series of the entire model domain. Representative Concentration Pathway (RCP) of the climate scenarios and Shelf Water (SW) or modified Circumpolar Deep Water (mCDW) ocean forcing shown in black (RCP4.5-SW), red (RCP8.5-mCDW), blue (RCP4.5-mCDW) and magenta (RCP8.5-mCDW). The error envelopes are determined from additional simulations for each climate scenario combining either the upper or lower estimates of both ocean and atmospheric forcing. Note that all results are shown relative to the reference run, and so are by definition 0 at the start of the model run.

scenario (Fig. 4a) and RCP8.5-SW scenario (Fig. 4b) both have a positive overall mass balance after 100 years, although increases in ice discharge have caused a reduction to +44 Gt a$^{-1}$ (−0.1 mm a$^{-1}$ SLE) and +143 Gt a$^{-1}$ (−0.4 mm a$^{-1}$ SLE), respectively. Individual regions maintain this trend of a positive mass balance that is smaller in magnitude than the first 100 years of simulation. The one exception is the D-D region (George V Land, including Cook and Ninnis Glaciers) in the RCP4.5-SW scenario which now has a negative mass balance of −14 Gt a$^{-1}$ in the RCP4.5-SW scenario, making it the only region to experience a negative mass balance change with SW forcing. Due to increased ice discharge over the final 100 years of model simulation the RCP4.5-mCDW scenario (Fig. 4:c) has a more negative mass balance of −241 Gt a$^{-1}$ (+0.7 mm a$^{-1}$ SLE) whilst the RCP8.5-mCDW scenario (Fig. 4d) has changed from a positive mass balance during the first 100 years of simulation to a negative mass balance of −132 Gt a$^{-1}$ (+0.4 mm a$^{-1}$ SLE) by the end of the simulation. The majority of regions in both simulations have a negative mass balance, with the exception of the D-E, A-B and the C-C region. A

**Table 1 | Summary of total ice volume, volume of ice above flotation (VAF), volume of floating ice and grounded area relative to the baseline case with constant forcing**

| | Total Volume (Gt) | | VAF (Gt) | | Floating ice (Gt) | | GA (km²) | |
|---|---|---|---|---|---|---|---|---|
| | 100 yrs | 200 yrs | 100 yrs | 200 yrs | 100 yrs | 200 yrs | 100 yrs | 200 yrs |
| RCP4.5-SW | 4824 | 12,120 | 4750 | 12,830 | −5323 | −10,295 | −595 | −4701 |
| RCP8.5-SW | 10,137 | 26,885 | 9963 | 26,730 | −6934 | −13,026 | 155 | −3835 |
| RCP4.5-CDW | −739 | −17,421 | −676 | −1718 | −40,568 | −74,440 | −16,956 | −47,174 |
| RCP8.5-CDW | 4573 | −1398 | 4526 | −1266 | −41,386 | −73,589 | −16,075 | −44,121 |

Results are shown for the four climate scenarios RCP4.5-SW, RCP8.5-SW, RCP4.5-mCDW and RCP8.5-mCDW (Representative Concentration Pathway, Shelf Water and Modified Circumpolar Deep Water respectively).

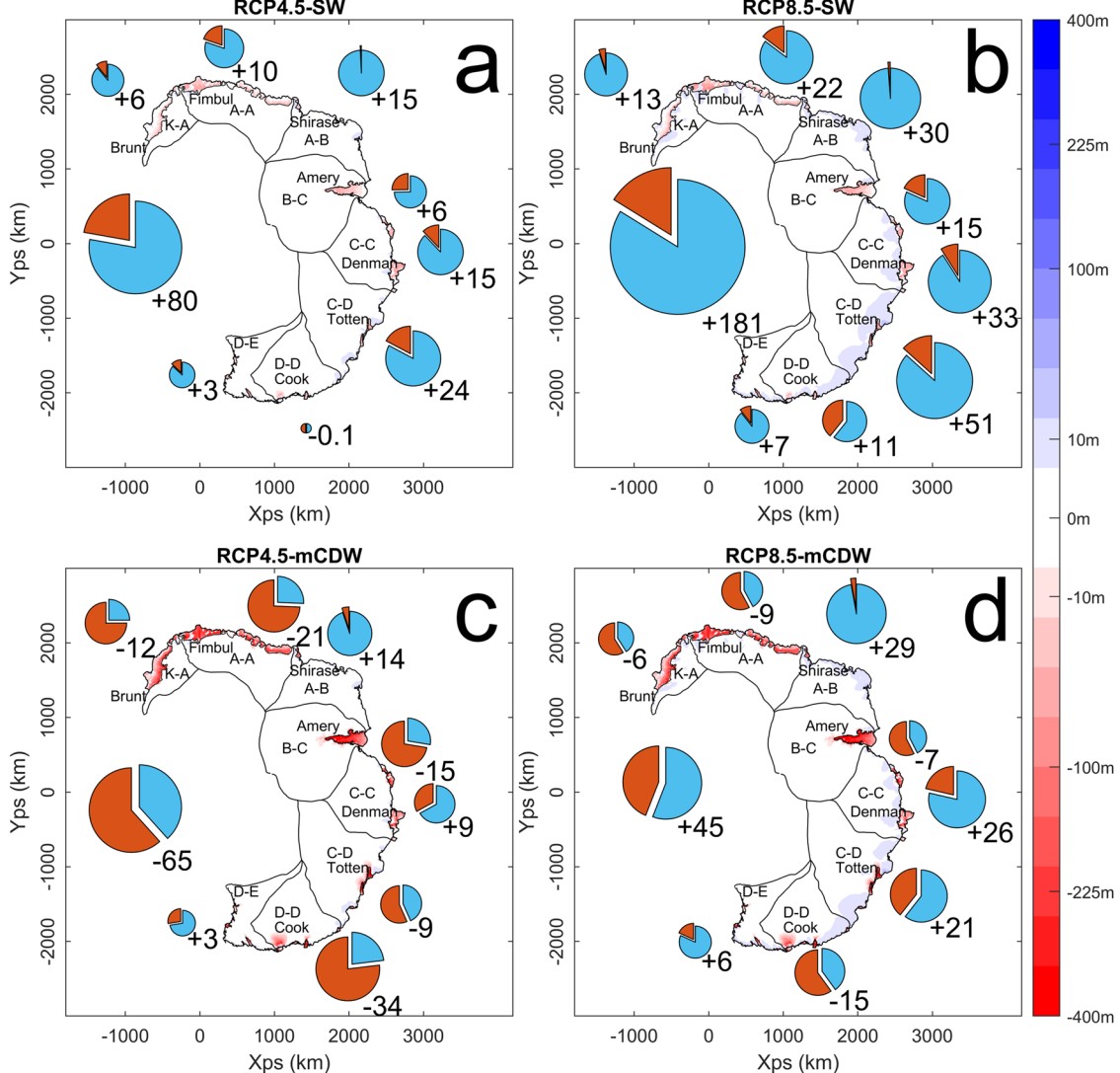

**Fig. 3 | Regional mass balance of the East Antarctic Ice Sheet after 100 years.** Total mass balance in GTa⁻¹ for the eight model catchments as well as the total model domain (pie charts, with area proportional to the net mass balance and the relative contribution of ice discharge in red and surface precipitation in blue) and ice thickness change (colours) after 100 years relative to the unforced scenario for **a** RCP4.5-SW, **b** RCP8.5-SW, **c** RCP4.5-mCDW and **d** RC8.5-mCDW climate scenarios (Representative Concentration Pathway, Shelf Water and Modified Circumpolar Deep Water respectively). Regions correspond to the IMBIE 2016 catchments[49].

summary of these results is given in Table 3, with in-depth results on a regional basis shown in the Supplementary material.

We can break down each regions response to ocean warming into three different categories (Fig. 5). The first, comprising of regions A–B, C–C and D–E, are regions that are always increasing in VAF under all emission scenarios. The second, comprising of regions K–A, A–A, B–C, and C–D, are regions that show a difference in response to the type of water mass driving ice shelf melting, with an increase in VAF with SW

forcing and a decrease in VAF with mCDW forcing. The final region, D–D, is a region that shows VAF decreasing by the end of the simulation for all emission scenarios except the RPC8.5-SW scenario, and is the only region experiencing a negative mass balance.

## Discussion

Our simulations show that if the cooler SW regime continues to be the primary driver of ocean-induced melting then the EAIS is likely to have

**Table 2 | Surface mass balance and ice discharge for the four climate forcing scenarios after 100 years of simulation time**

| Region | RCP4.5-SW | | RCP8.5-SW | | RCP4.5-mCDW | | RCP8.5-mCDW | |
|---|---|---|---|---|---|---|---|---|
| | SMB (Gt) | D (Gt) | SMB (Gt) | D (Gt) | SMB (Gt) | D (Gt) | SMB (Gt) | D (Gt) |
| K–A | 7 | 1 | 14 | 1 | 6 | 18 | 13 | 19 |
| A–A | 12 | 3 | 26 | 4 | 11 | 32 | 24 | 33 |
| A–B | 15 | 0 | 30 | 0 | 15 | 1 | 30 | 1 |
| B-C | 9 | 3 | 19 | 4 | 9 | 24 | 18 | 25 |
| C–C | 18 | 3 | 37 | 4 | 18 | 9 | 36 | 10 |
| C–D | 30 | 6 | 60 | 9 | 29 | 38 | 59 | 38 |
| D–D | 15 | 15 | 31 | 20 | 14 | 48 | 30 | 45 |
| D–E | 4 | 1 | 8 | 1 | 4 | 1 | 8 | 2 |
| Total | 112 | 32 | 224 | 43 | 106 | 171 | 217 | 172 |

Results are shown for the four climate scenarios RCP4.5-SW, RCP8.5-SW, RCP4.5-mCDW and RCP8.5-mCDW (Representative Concentration Pathway, Shelf Water and Modified Circumpolar Deep Water respectively).

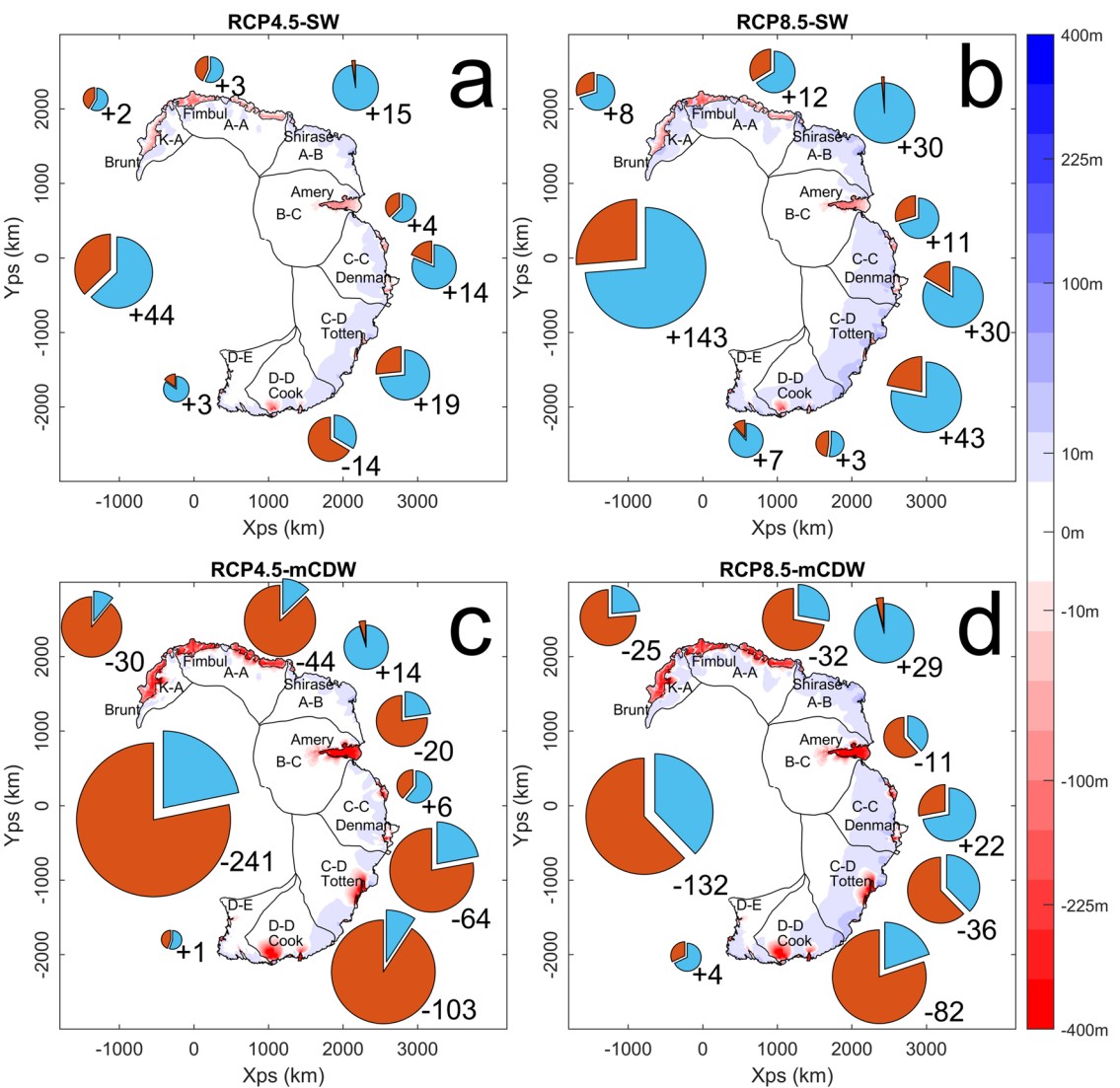

**Fig. 4 | Regional mass balance of the East Antarctic Ice Sheet after 200 years.** Total mass balance in GTa$^{-1}$ for the eight model catchments as well as the total model domain (pie charts, with area proportional to the net mass balance and the relative contribution of ice discharge in red and surface precipitation in blue) and ice thickness change (colours) after 200 years relative to the unforced scenario for **a** RCP4.5-SW, **b** RCP8.5-SW, **c** RCP4.5-mCDW and **d** RC8.5-mCDW climate scenarios (Representative Concentration Pathway, Shelf Water and Modified Circumpolar Deep Water respectively). Regions correspond to the IMBIE 2016 catchments[49].

**Table 3 | Surface mass balance and ice discharge for the four climate forcing scenarios after 200 years of simulation time**

| Region | RCP4.5-SW | | RCP8.5-SW | | RCP4.5-mCDW | | RCP8.5-mCDW | |
|---|---|---|---|---|---|---|---|---|
| | SMB (Gt) | D (Gt) | SMB (Gt) | D (Gt) | SMB (Gt) | D (Gt) | SMB (Gt) | D (Gt) |
| K–A | 7 | 5 | 14 | 6 | 4 | 34 | 11 | 36 |
| A–A | 12 | 9 | 25 | 12 | 8 | 52 | 20 | 53 |
| A–B | 15 | 0 | 30 | 0 | 15 | 1 | 30 | 1 |
| B–C | 9 | 6 | 18 | 8 | 8 | 28 | 18 | 28 |
| C–C | 18 | 4 | 37 | 7 | 17 | 11 | 36 | 14 |
| C–D | 30 | 11 | 61 | 17 | 26 | 90 | 56 | 93 |
| D–D | 15 | 29 | 30 | 27 | 12 | 116 | 27 | 110 |
| D–E | 4 | 1 | 8 | 1 | 4 | 3 | 8 | 4 |
| Total | 109 | 65 | 222 | 79 | 93 | 335 | 205 | 338 |

Results are shown for the four climate scenarios RCP4.5-SW, RCP8.5-SW, RCP4.5-mCDW and RCP8.5-mCDW (Representative Concentration Pathway, Shelf Water and Modified Circumpolar Deep Water respectively).

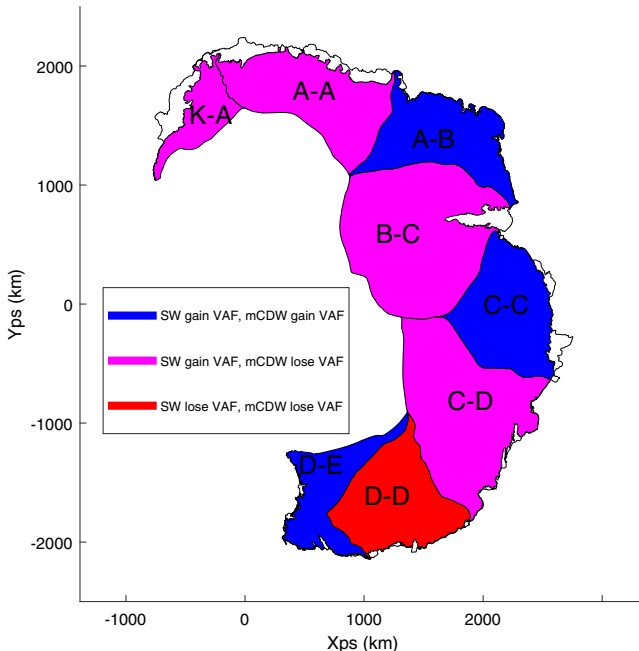

**Fig. 5 | Susceptibility of East Antarctic catchments to ocean warming over the next 200 years.** Blue regions show increased VAF (Volume Above Floatation) under all forcing scenarios, magenta regions show increased VAF for SW (Shelf Water) forced scenarios and decreased VAF for mCDW (modified Circumpolar Deep Water forced scenarios whilst red regions show reduced VAF for all scenario except the RCP8.5-SW (Representative Concentration Pathway). Regions correspond to the IMBIE 2016 catchments[49].

floating ice as well as grounded area, even when they show an increase in VAF or total ice volume. If this were to continue into the future, East Antarctic ice shelves are likely to be greatly reduced in both thickness and extent compared to their current configurations. Simulations that directly impose an instantaneous collapse of East Antarctic ice shelves have found this to lead to the destabilisation of the Wilkes and Aurora Subglacial Basins over the next 500 years[40]. More transient simulations of the EAIS find that the removal of grounded ice or atmospheric driven surface melting are required to make the EAIS susceptible to collapse, with only the Recovery catchment in the Weddell Sea region (an area that lies outside of our model domain) proving susceptible to ocean driven collapse[41]. Model simulations of the Wilkes Subglacial Basin have found it to be stable in response to increased ocean warming, however if this increased ocean driven melt were to remove a plug of ice then the Wilkes Basin would become destabilised[42].

In contrast, if the water masses around East Antarctica were to switch to a mode that is dominated by enhanced intrusion of mCDW, our simulations reveal that the EAIS will have a negative overall change in VAF after the 200 years of model simulations for both the RCP4.5-mCDW (−17,421 Gt,+48 mm SLE) and the RCP8.5-mCDW (−1398 Gt, +4 mm SLE) scenarios. Whilst the RCP8.5-mCDW scenario has a positive mass balance after 100 years this becomes negative after 200 years due to maximum ice discharge lagging behind the application of maximum oceanic forcing at year 100. The RCP4.5-mCDW scenario has a negative mass balance for the entirety of the simulation, as the relative increase of ice discharge compared to precipitation is greater throughout the entire 200 years of simulation time. Our simulations show that a switch to mCDW becoming the primary water mass driving ice-shelf melting is sufficient to more than counteract any increased mass gained from precipitation, leading to positive sea level rise contribution from the EAIS over the next 200 years (+4–48 mm SLE contribution). The reduction in both floating ice and grounded area is more pronounced in both the mCDW forced climate scenarios than in the SW forced scenarios, indicating an increased likelihood of ice shelf collapse and run away grounding line retreat that might occur in catchments with reverse-sloping beds inland of present grounding lines. Paleoenvironmental and paleoceanographic records records in the Amundsen sea sector[43] and Wilkes Subglacial Basin[31,32] show this process has occurred in the past, linking indicators of mCDW intrusions with ice-shelf retreat.

Our simulations consistently find that region D-D, and in particular Cook and Ninnis Glacier, is the most susceptible to ocean driven increased ice discharge. It is the only region to show a negative mass balance under an SW forced climate scenario, losing −14 Gt a$^{-1}$ in the RCP4.5-SW scenario after 200 years. It is also the most strongly affected region in the mCDW forced scenarios and the susceptibility of this region to both past[31,32] and future[42,44] climatic shifts has been highlighted by other studies. Findings from the ISMIP6 project[5] found

gained VAF after 200 years for both the RCP4.5-SW (+12,120 Gt,−33 mm SLE) and RCP8.5-SW (+26,885 Gt, −74 mm SLE) scenarios. This is in broad agreement with the ISMIP6 results[5] which found a majority of models predicting a positive mass balance for the EAIS as well as other estimates for future East Antarctic mass balance[8,37,38]. Pollard and DeConto[39] find Totten glacier (basin C–D) beginning to undergo severe retreat and thinning by 2100, with our simulations only showing comparable results for the mCDW driven scenarios. Their model framework includes parameterisations of surface melt induced hydro-fracturing of ice shelves as well as marine ice cliff instability that our model framework does not, perhaps explaining the discrepancy. We find that the mass gain from increased precipitation outweighs the mass loss from increased ice discharge due to ice-shelf melt in SW forced simulations. It should be noted, however, that this is not necessarily a steady state. All our simulations show a reduction in

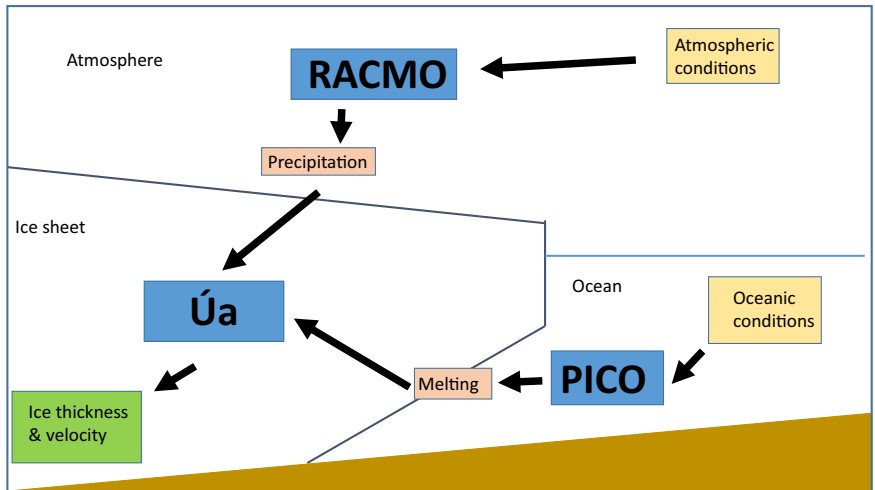

**Fig. 6 | Schematic set up of model framework.** The ice dynamics model Úa is forced with oceanic melt rates derived by the PICO (Potsdam Ice-shelf Cavity mOdel) box model and RACMO (Regional Atmospheric Climate Model) precipitation. PICO uses an ensemble of CMIP5 model mean ocean properties to derive its melt rates, whilst the present day RACMO precipitation is scaled in line with projected CMIP5 (Climate Model Intercomparison Project) model mean temperature increases. Úa determines future ice thickness and velocity which can the be used to determine overall mass balance.

Wilkes Land to be the most strongly affected region of East Antarctica, with Totten Glacier to be particularly susceptible to future climatic warming, albeit with a spread of results that our simulations fall within. Our results show a thinning of the ice sheet interior in George V land that we do not see in Wilkes land leading to a more muted effect on sea level contribution from Wilkes Land in our results despite a comparable, albeit slightly greater, reduction in grounded area (a good approximation to grounding line retreat) in George V Land compared to Wilkes land. In contrast, we find that region A–B, C–C and D–E to be least affected by increased oceanic melting. Indeed, these regions are the only regions to show a consistently positive mass balance throughout all climate scenarios. This is most likely a result of a lack of large ice shelves in theses regions combined with the prevalence of inland, terrestrial ice limiting their susceptibility to warming ocean conditions. In contrast, multi-millennial simulations (albeit with a coarse grid resolution limiting grounding line migration) for the next 10,000 years of the EAIS[41] have found that the entirety of the EAIS covered by our domain is negligibly affected by ocean warming of a comparable forcing to our mCDW scenarios whilst Recovery basin (outside our model domain) was strongly affected by a warming Weddell Sea. When subjected to atmospheric warming of 8 °C (greater than the 4 °C warming assumed by our RCP8.5 scenarios) they find that Wilkes Land and George V Land were the most susceptible to surface melting, a process our model framework does not include.

Our results clearly show that there is a lag in the maximum of ice discharge relative to increased oceanic melting, with an increase in the ice discharge over the second half of the simulations when the maximum temperature forcing is held constant. This is similar to the results of Lowry et al.[45] that highlight the potential for increased sea level contribution beyond 2100 from the EAIS. This is in broad agreement with a recent review of future EAIS mass balance predictions out to 2500[9] that found a greater rate of sea level contribution in the period after 2100 than before it, albeit with correspondingly larger errors. In our results, for a given increase in oceanic melting there is an acceleration in ice discharge rather than an instantaneous increase to a greater velocity. This is demonstrated, particularly for the mCDW forced scenarios, by the rate of change in total ice volume (Fig. 2b) being primarily forced by the change in floating ice (Fig. 2d) for the first hundred years whilst over the second 100 years of the simulation VAF loss increase to a comparable size to floating ice loss. This is most pronounced for the C–D region, which drains the Aurora subglacial basin, which has a positive mass balance in the RCP8.5-mCDW scenario

after 100 years of simulation time and a negative mass balance after 200 years due to a more than doubling of its ice discharge in the last 100 years of simulated run time. We also find that our results consistently show the most negative mass balance arising from the intermediate warming RCP4.5-mCDW scenario rather than the extreme RCP8.5-mCDW scenario. Other work on East Antarctic mass balance has found similar results, with predicted 2100 mass balance being dominated by precipitation rather than ocean melting, showing a more positive trend with increased warming[37]. This can be explained by there being a relatively larger increase in precipitation between the RCP4.5 and RCP8.5 simulations than there is in ice discharge.

Our simulations of future East Antarctic mass balance highlight the potential importance of indirect effects arising from global heating. In our particular case, in addition to a general warming of the atmosphere and ocean, global heating has the potential to shift oceanic circulation in the Southern Ocean, changing the water mass in contact with Antarctic ice shelves from the relatively cool Shelf Water regime seen today to a warmer regime dominated by intrusions of Circumpolar Deep Water. This leads to a >400% increase in the amount of ice discharge in the RCP4.5 and RCP8.5 simulations forced by modified Circumpolar Deep Water when compared to the RCP4.5 and RCP8.5 simulations forced by Shelf Water. The increase in ice discharge between RCP4.5 and RCP8.5 Shelf Water forced scenarios, in contrast, is only 25% whilst the difference between RCP4.5 and RCP8.5 modified Circumpolar Deep Water is a negligible 0.5%. Whilst we are not suggesting the entirety of East Antarctica is likely to switch to our worst-case scenario with melting dominated by modified Circumpolar Deep Water in all regions, the possibility of increased modified Circumpolar Deep Water intrusions (particularly in areas of high risk, such as George V Land) can have a large potential contribution to sea level rise from East Antarctica over the next couple of hundred years.

## Methods
We conduct experiments using the finite element, ice dynamics model Úa[46], which has previously been used for large scale, transient simulations of Antarctic future mass balance[47] as well as smaller, more localised studies such as an investigation into the past behaviour of Cook glacier[44] or rift propagation on the Brunt Ice Shelf[48]. The model domain has been deliberately chosen to encompass the drainage basins corresponding to the IMBIE 2016 definition of East Antarctica[49], with the exception of the J'-K and E-E' basins which drain into the Filchner-Ronne and Ross ice shelves, respectively. Our work chooses to

focus on the potential for future mass loss from the comparatively less studied parts of the EAIS. Including these basins would also have necessitated the inclusion of much of West Antarctica to fully model the Ross and Filchner Ronne ice shelves, thus drastically increasing the computational expense of our simulations. The resulting model domain is shown in Fig. 1. We derive ocean melt rates via an implementation of the PICO (Potsdam Ice-shelf Cavity mOdel[50]), similar to that used by Hill et. al.[38] whilst precipitation is provided by the RACMO (Regional Atmospheric Climate Model) version 2.3 data set[51]. A schematic representation of the model framework is shown in Fig. 6.

Both increasing ocean melt and surface precipitation are driven by a single representative temperature values that increase to represent future climate scenarios, with the increase derived from analysis of the model mean CMIP5[52] results. Each of our future climate scenarios assumes a linearly increasing temperature which reaches a maximum 100 years from the present. This new maximum is then held constant for a further 100 years to explore the committed or long-term response to the first 100 years of forcing. Ice thickness and velocity are then used to determine ice discharge, which is combined with surface precipitation to give the overall mass balance, defined here as

$$MB = P - D \qquad (1)$$

where $MB$ is the mass balance, $P$ the surface precipitation and $D$ the ice discharge over the grounding line. A positive $MB$ indicates that the EAIS is gaining mass whilst a negative $MB$ indicates that the EAIS is losing mass. Note that all results are shown in reference to a baseline simulation with constant forcing representing present day conditions. This has the effect of limiting model bias during initialisation, with results showing the relative effect of imposing a given climate scenario rather than absolute values.

We consider our choice of single, representative values for both the increase in oceanic and atmospheric forcing to be valid for the purpose of comparing and isolating the relative impacts on EAIS mass balance of a shift in ocean forcing from the relatively cool, current regime dominated by shelf water to a warmer regime dominated by modified Circumpolar Deep Water. As such our results should be considered in the context of what is likely to happen if modified circumpolar deep water dominates ice shelf melting rather than an exact representation of a particular simulation of a future climate scenario from an individual CMIP model. A more in depth description of the model framework, calibration and forcing can be found in the accompanying Supplementary material.

## Data availability
Model data is archived with the United Kingdoms Polar Data Centre and is freely available from https://doi.org/10.5285/BEDA45D1-DD33-4666-8861-B4B91AF0180F[53].

## Code availability
Simulations were performed using the finite element, ice dynamics model Úa[46], code available from https://github.com/GHilmarG/UaSource.

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

## Acknowledgements
This work was funded by the Natural Environment Research Council, grant number NE/R000719/1. This publication was supported by PROTECT. J.R.J., H.G. and A.J. have received funding from the European Union's Horizon 2020 research and innovation program under grant agreement No 869304, PROTECT contribution number 55. B.W.J.M. was supported by a Leverhulme Early Career Fellowship (ECF-2021-484).

## Author contributions
J.R.J. designed the experiments, performed the numerical simulations, conducted the analysis, made the figures and led the manuscript writing. B.W.J.M. assisted in the making of the figures. All authors (J.R.J., B.W.J.M., C.R.S., G.H.G., S.S.R.J. and A.J.) provided input on the interpretation and discussion of results and commented on the manuscript.

## Competing interests
The authors declare no competing interests.
