## [Peer review file · Nature Communications]

Increased warm water intrusions could cause mass loss in East Antarctica within 200 yearsEditorial Note: This manuscript has been previously reviewed at another journal that is not operating a transparent peer review scheme. This document only contains reviewer comments and rebuttal letters for versions considered at *Nature Communications*.

Increased warm water intrusions could cause mass loss in East Antarctica within 200 years" by Jordan et al. (NCLIM-22040671)

We thank the reviewers for their comments and feedback and attach a revised version of the manuscript. Due to combined figure and table limits the majority of the methods section has now been moved to the supplementary material. In the following detailed responses, the reviewers' comments are shown in blue with our replies in black.

Reviewer #1 (Remarks to the Author):

Review of Jordan et al, "Increased warm water intrusions could cause mass loss in East Antarctica within 200 years'

The manuscript presents a suite of ice sheet model simulations that explore the possible future response of the East Antarctic Ice Sheet to future environmental conditions. Loosely framed in terms of two emissions scenarios, the experiments show that it is only if the effects of upwelling circumpolar deep water are included that the EAIS loses mass. This means that high latitude ocean circulatory changes are more important than changes in the global mean climate state. To my mind, this is the most important finding of the study, and yet I don't feel like the paper currently emphasises this aspect sufficiently clearly (see comments below). That aside, the paper is improved from the version I previously reviewed for Nat. Clim. Change, and I still think the figures are great and present the results very clearly. The addition of the new figures is also appreciated.

I think it's a valuable paper and I only have a few comments further to those made previously

Line numbers refer to those in the track-changes pdf

l1-2 - I still have a problem with this title, it really undersells the paper. It's a no-brainer that warm water would cause mass loss, so I think the strength / novelty of the paper is in showing that ocean circulation NOT emissions scenarios per se are the critical thing for EAIS future. What about a title that reflects that distinction, e.g. "Future East Antarctic mass loss controlled more by ocean circulation change than climate trajectory" or something like that.

We agree that the potential impact on EAIS mass loss from a change in ocean circulation is the main finding from our work, but this change in ocean circulation only arises due to a warming climate trajectory and does not occur independently from it, which could be implied by the suggested title. We feel that our current title is the best representation of the manuscripts contents.

l20 - 'after 200 years' - from the figures it seems that the NMB is -ve throughout the run under the mCDW-intrusion conditions

Sentence has been changed to reflect the -ve mass balance over the whole simulation run time, rather than just at the end.

l27 - 'greater' - I find this ambiguous, because the terms have different signs. I think it might be clearer to say 'more negative'

Sentence has been changed to "more negative" for increased clarity.

l91 - 'Paleo records' - could be more specific - 'paleoenvironmental', 'paleoceanographic', etc...

Changed to “Paleoenvironmental and paleoceanographic records” to be more specific as to the nature of the records.

l103-5 - in line with the suggested change in the title, you could say here that you're trying to quantify the differing consequences of ocean circ vs emission scenario

Sentence has been changed to state our aim more explicitly in showing the consequences of a shift in oceanic circulation.

l228 - 'then'  'than'

Changed.

l356 - 'Paleo records' - see previous comment

Changed in line with previous comment.

l365, 369, 377, 384, 386 - 'effected'  'affected'

Changed.

Reviewer #2 (Remarks to the Author):

The authors have satisfactorily responded to all of my major comments except for number 3. regarding the expected projections for the Antarctic Slope Current.

The authors are correct that a shift to positive SAM would result in weakened easterlies and therefore a reduced ASC. However, as Beadling et al 2022 show in their CM4 simulation, this response is projected to be overwhelmingly dominated by an opposing response, i.e. an increase in the ASC, driven by the additional meltwater over the shelf (see their Figure 10a). The authors' current statement in the conclusions is at odds with our best understanding of the future response of the ASC, i.e. lines 389-391: "has the potential to reduce the strength of the Antarctic Slope Front due to a more positive Southern Annular Mode, thus presenting less of a barrier to warm modified Circumpolar Deep Water intrusions".

I don't see this as a problem for the authors' argument that there may be ocean warming on the continental shelf in the future, as ocean warming is possible even with an increased ASC, e.g. as shown in Beadling et al. Figure 6. There are more components to the ocean heat budget of the continental shelf than just the advective flux across the continental slope (e.g. as explained in the Ross Sea heat budget of Moorman et al. 2020 - ocean shelf temperatures increase even as the cross slope heat transport reduces, see their Figure 10). This just requires some rewording, so that the authors are not implying that we expect the ASC to reduce in the future.

We thank the reviewer for their insight and feedback on this topic and have adjusted the sentence in question to refer to CDW intrusions being more likely in a warming climate.

REVIEWERS' COMMENTS

Reviewer #1 (Remarks to the Author):

Review of Jordan et al, "Increased warm water intrusions could cause mass loss in East Antarctica within 200 years"

The manuscript presents a suite of ice sheet model simulations that explore the possible future response of the East Antarctic Ice Sheet to future environmental conditions. Loosely framed in terms of two emissions scenarios, the experiments show that it is only if the effects of upwelling circumpolar deep water are included that the EAIS loses mass. This means that high latitude ocean circulatory changes are more important than changes in the global mean climate state. To my mind, this is the most important finding of the study, and yet I don't feel like the paper currently emphasises this aspect sufficiently clearly (see comments below). That aside, the paper is improved from the version I previously reviewed for Nat. Clim. Change, and I still think the figures are great and present the results very clearly. The addition of the new figures is also appreciated.

I think it's a valuable paper and I only have a few comments further to those made previously

Line numbers refer to those in the track-changes pdf

I1-2 - I still have a problem with this title, it really undersells the paper. It's a no-brainer that warm water would cause mass loss, so I think the strength / novelty of the paper is in showing that ocean circulation NOT emissions scenarios per se are the critical thing for EAIS future. What about a title that reflects that distinction, e.g. "Future East Antarctic mass loss controlled more by ocean circulation change than climate trajectory" or something like that.

I20 - 'after 200 years' - from the figures it seems that the NMB is -ve throughout the run under the mCDW-intrusion conditions

I27 - 'greater' - I find this ambiguous, because the terms have different signs. I think it might be clearer to say 'more negative'

I91 - 'Paleo records' - could be more specific - 'paleoenvironmental', 'paleoceanographic', etc...

I103-5 - in line with the suggested change in the title, you could say here that you're trying to quantify the differing consequences of ocean circ vs emission scenario

I228 - 'then'  'than'

I356 - 'Paleo records' - see previous comment

I365, 369, 377, 384, 386 - 'effected'  'affected'

Reviewer #2 (Remarks to the Author):

The authors have satisfactorily responded to all of my major comments except for number 3. regarding the expected projections for the Antarctic Slope Current.

The authors are correct that a shift to positive SAM would result in weakened easterlies and therefore a reduced ASC. However, as Beadling et al 2022 show in their CM4 simulation, this response is projected to be overwhelmingly dominated by an opposing response, i.e. an increase in the ASC, driven by the additional meltwater over the shelf (see their Figure 10a). The authors' current statement in the conclusions is at odds with our best understanding of the future response of the ASC, i.e. lines 389-391: "has the potential to reduce the strength of the Antarctic Slope

Front due to a more positive Southern Annular Mode, thus presenting less of a barrier to warm modified Circumpolar Deep Water intrusions".

I don't see this as a problem for the authors' argument that there may be ocean warming on the continental shelf in the future, as ocean warming is possible even with an increased ASC, e.g. as shown in Beadling et al. Figure 6. There are more components to the ocean heat budget of the continental shelf than just the advective flux across the continental slope (e.g. as explained in the Ross Sea heat budget of Moorman et al. 2020 - ocean shelf temperatures increase even as the cross slope heat transport reduces, see their Figure 10). This just requires some rewording, so that the authors are not implying that we expect the ASC to reduce in the future.